# Prognostic Factors for Pulmonary Fibrosis Following Pneumonia in Patients with COVID-19: A Prospective Study

**DOI:** 10.3390/jcm11195913

**Published:** 2022-10-07

**Authors:** Inhan Lee, Joohae Kim, Yohwan Yeo, Ji Yeon Lee, Ina Jeong, Joon-Sung Joh, Gayeon Kim, Bum Sik Chin, Yeonjae Kim, Min-Kyung Kim, Jaehyun Jeon, Yup Yoon, Sung Chan Jin, Junghyun Kim

**Affiliations:** 1Division of Pulmonary and Critical Care Medicine, Department of Internal Medicine, National Medical Center, Seoul 04564, Korea; 2Department of Family Medicine, College of Medicine, Hallym University Dongtan Sacred Heart Hospital, Hwaseong 18450, Korea; 3Division of Infectious Diseases, Department of Internal Medicine, National Medical Center, Seoul 04564, Korea; 4Department of Radiology, National Medical Center, Seoul 04564, Korea; 5Division of Pulmonary, Allergy and Critical Care Medicine, Department of Internal Medicine, Hallym University Dongtan Sacred Heart Hospital, Hallym University College of Medicine, Hwaseong 18450, Korea

**Keywords:** pulmonary fibrosis, COVID-19, pneumonia

## Abstract

The frequency and clinical manifestation of lung fibrosis accompanied by coronavirus disease (COVID-19) are not well-established. We aimed to identify the factors attributed to post-COVID-19 fibrosis. This single-center prospective study included patients diagnosed with COVID-19 pneumonia from 12 April to 22 October 2021 in the Republic of Korea. The primary outcome was the presence of pulmonary fibrosis on a CT scan 3 months after discharge; the fibrosis risk was estimated by a multiple logistic regression. The mean patient age was 55.03 ± 12.32 (range 27–85) years; 65 (66.3%) were men and 33 (33.7%) were women. The age, Charlson Comorbidity Index, lactate dehydrogenase level, aspartate aminotransferase level, and Krebs von den Lungen-6 level were significantly higher and the albumin level and the saturation of the peripheral oxygen/fraction of inspired oxygen (SpO_2_/FiO_2_) ratio were significantly lower in the fibrosis group than in the non-fibrosis group; the need for initial oxygen support was also greater in the fibrosis group. An older age (adjusted odds ratio (AOR) 1.12; 95% confidence interval (CI) 1.03–1.21) and a lower initial SpO_2_/FiO_2_ ratio (AOR 7.17; 95% CI 1.72–29.91) were significant independent risk factors for pulmonary fibrosis after COVID-19 pneumonia. An older age and a low initial SpO_2_/FiO_2_ ratio were crucial in predicting pulmonary fibrosis after COVID-19 pneumonia.

## 1. Introduction

The first case of infection due to severe acute respiratory syndrome coronavirus 2 (SARS-CoV-2) was reported in Wuhan, China, in December 2019. The infection was subsequently named coronavirus disease (COVID-19) by the World Health Organization (WHO). Since then, the COVID-19 pandemic has persisted as a global infection [1,2,3], with the number of confirmed cases surpassing half a billion and six million deaths to date.

Many studies have reported at least one persistent symptom in patients after their recovery from COVID-19. Post-COVID-19 symptoms range from respiratory symptoms such as exertional dyspnea, chest discomfort, and a cough to non-respiratory symptoms such as fatigue, cognitive impairments, and psychological problems such as anxiety and depression [4,5,6].

In patients hospitalized for COVID-19 pneumonia, a ground-glass opacity pattern is a typical initial radiologic finding and the pattern varies from severe acute respiratory distress syndrome to interstitial pneumonia [7,8]. Studies have reported that in moderate to severe COVID-19 pneumonia, the typical ground-glass opacification patterns turn into fibrotic changes, causing residual respiratory symptoms such as dyspnea and a cough for months [9,10]. Studies on whether the sequelae of COVID-19 pneumonia persist in certain patients, especially survivors, are in progress. 

Currently, an antiviral therapy, an anti-inflammatory therapy (corticosteroids), and supportive care are the main pillars of COVID-19 pneumonia treatment; no specific preventive therapy or treatment has been officially recommended for pulmonary fibrosis. Clinical trials, including randomized controlled trials (ClinicalTrials.gov identifier: NCT04541680, NCT04619680, and NCT04607928), are investigating the possibility of applying antifibrotic agents such as pirfenidone and nintedanib, which have already been proven to be effective for treating idiopathic pulmonary fibrosis [11,12]. Due to the increasing number of cases, survivors need further attention and the long-term changes in COVID-19-infected lungs and symptoms must be studied.

Thus, we aimed to identify the factors attributed to post-COVID-19 fibrosis to provide a better understanding of the disease entity and additional treatment options for survivors of COVID-19. In this study, we analyzed 3-month follow-up CT scans to evaluate the fibrotic progression and its risk factors in patients who had been diagnosed with COVID-19.

## 2. Materials and Methods

### 2.1. Study Design and Population

This single-center prospective study included patients who were diagnosed with COVID-19 pneumonia. The entire population of 519 patients hospitalized at the National Medical Center in the Republic of Korea from 12 April to 22 October 2021 was screened. The patient inclusion criteria were as follows: (1) age > 18 years; (2) SARS-CoV-2 infection confirmed by a positive nasopharyngeal swab reverse transcriptase-polymerase chain reaction; and (3) a diagnosis of pneumonia from an initial chest X-ray. Patients who refused to consent to this study, were unable to participate, were unable to express their will to consent due to unconsciousness and the absence of a legal deputy, or were unable to return to the outpatient clinic for a follow-up chest CT scan 3 months after the hospital discharge were excluded (Figure 1).

### 2.2. Study Variables

All information was collected from electronic medical records. Demographic characteristics, including age, sex, body mass index (BMI), smoking history, and underlying comorbidities, were included in the analysis. For the clinical status at admission, the vital signs, oxygen support, laboratory results, and Eastern Cooperative Oncology Group performance status were reviewed. During hospitalization, the variables related to the hospital course such as the length of hospital stay, intensive care unit admission, maximum oxygen support, and medication were also collected. Medications included antiviral agents (remdesivir), monoclonal antibodies (regdanvimab and tocilizumab), steroids, and antibacterial agents. Antifibrotic agents that did not have clear evidence [13] of use and were not approved by the Korean Food and Drug Administration (FDA) were not applied to the study population.

Among laboratory findings, Krebs von den Lungen-6 (KL-6), a serological biomarker for interstitial lung disease, has the potential to predict fibrosis in patients with COVID-19 pneumonia [14,15]. In our study, however, we started to collect KL-6 from the middle of the study period; thus, data were only available for 79 patients. We referred to multiple studies published in 2021 that reported a correlation between KL-6 and post-COVID-19 fibrosis [14,16].

### 2.3. Study Outcome

The primary outcome of this study was the presence of pulmonary fibrosis on CT scans 3 months after discharge. Two independent radiologists specializing in pulmonology radiologic imaging were fully blinded to any other clinical findings and evaluated the CT scans for the presence of fibrosis. Pulmonary fibrosis was defined when any of the following radiologic features were present: (1) parenchymal bands; (2) traction bronchiectasis with or without volume loss; (3) reticulation; and (4) honeycombing [17,18,19]. In the case of a discrepancy between the assessments of the two radiologists, a final consensus was reached after a discussion. The inter-rater reliability between the radiologists was substantial, with a Cohen’s kappa of 0.938.

The study protocol was approved by the Institutional Review Board of the National Medical Center (NMC-2021-04-032). The present study was registered with the Clinical Research Information Service (No. KCT0006312). Written informed consent was provided by individual patients or their caregivers with a legal deputy. 

### 2.4. Statistical Analyses

The descriptive statistics were presented as numbers (percentages) for the categorical variables and as the mean ± standard deviation for the continuous variables. We compared the two groups using the χ^2^ or Fisher’s exact test for the categorical variables, as appropriate, and the Student’s *t*-test for the continuous variables. The risk of post-COVID-19 pulmonary fibrosis was estimated for each potential risk factor (age, sex, BMI, and smoking status) and the variables having *p*-values < 0.10 were included in the multiple logistic regression models. Adjusted odds ratios (ORs) and 95% confidence intervals (CIs) were calculated and a *p*-value < 0.05 was considered to be statistically significant. All statistical analyses were performed using the SAS statistical package (version 9.4; SAS Institute Inc., Cary, NC, USA).

## 3. Results

### 3.1. Baseline Characteristics

Among the 124 patients initially enrolled in this study, 5 died during admission, 12 refused follow-up visits, and 9 were lost to follow-up (Figure 1). A total of 98 patients who underwent a CT scan follow-up 3 months after discharge were included. Pulmonary fibrosis was observed in 43 (43.9%) patients.

Table 1 shows the clinical characteristics of the study population. The mean age was 55.03 ± 12.32 (range 27–85) years; 65 (66.3%) of the patients were men and 33 (33.7%) were women. The age, Charlson Comorbidity Index, lactate dehydrogenase (LDH) level, aspartate aminotransferase (AST) level, and KL-6 level were significantly higher and the albumin level and the saturation of the peripheral oxygen/fraction of inspired oxygen (SpO_2_/FiO_2_) ratio were significantly lower in the fibrosis than in the non-fibrosis group; the need for initial oxygen support was also greater in the fibrosis group. 

The clinical courses of these patients are summarized in Table 2. More patients in the fibrosis group (*N* = 19, 44.2%) were admitted to the intensive care unit compared with the non-fibrosis group (*N* = 10, 18.2%) (*p* = 0.005). A higher proportion of patients in the fibrosis group than in the non-fibrosis group required an oxygen supply during the entire hospital stay and at a higher concentration of oxygen (*p* = 0.009). The rate of antiviral agent and steroid use was also higher in the fibrosis group (*N* = 37, 86.0% for the antiviral agent and *N* = 41, 95.3% for the steroids) compared with the non-fibrosis group (*N* = 33, 60.0% for the antiviral agent and *N* = 37, 69.8% for steroids).

### 3.2. Risk Factors for Post-COVID-19 Pulmonary Fibrosis

In the multivariate regression analysis, an older age (adjusted odds ratio (AOR) 1.12; 95% CI 1.03–1.21) and a lower initial SpO_2_/FiO_2_ ratio (AOR 7.17; 95% CI 1.72–29.91) were found to be significant independent risk factors for pulmonary fibrosis after COVID-19 pneumonia (Table 3).

### 3.3. Characteristics of Radiological Findings in the Fibrosis Group

The detailed radiologic features of the 43 follow-up chest CT scans from the fibrosis group 3 months after discharge are summarized in Table 4. Parenchymal bands were present in all CT scans, with evidence of pulmonary fibrosis this accounted for 43.88% of the total study population. Other fibrotic features were also observed in relatively small populations; traction bronchiectasis with or without volume loss was present in eight (8.16%), reticulation in one (1.02%), and honeycombing in two (2.04%) follow-up CT scans.

## 4. Discussion

This study was conducted in a single national hospital to identify the clinical courses and prognostic factors of pulmonary fibrosis after COVID-19 pneumonia by investigating patients who developed fibrotic sequelae in the 3-month follow-up CT scans. A total of 43 (43.9%) out of 98 patients presented with a fibrotic change. An older age and a low initial SpO_2_/FiO_2_ ratio were significant risk factors for pulmonary fibrosis after COVID-19. In the chest CT findings, parenchymal bands were the dominant feature in the fibrosis group; traction bronchiectasis with/without volume loss, reticulation, or honeycombing appeared in less than one-tenth of the population.

The significant prognostic factors identified in our study were consistent with the findings of other studies. An older age is a widely acknowledged risk factor for pulmonary fibrosis after COVID-19 in many multivariate analyses [20,21,22]. In a Chinese study, although 61% of the study population presented a complete radiological resolution after seven months, older patients were more likely to develop fibrosis [23]. In previous outbreaks of coronaviruses such as severe acute respiratory syndrome and Middle East respiratory syndrome coronavirus, an older age has also been suggested to be a risk factor for pulmonary fibrosis [24,25]. Fibrotic changes after a respiratory viral infection could be explained by the increasing disease severity in older patients or physiological susceptibility of the aged lung [26].

A lower initial SpO_2_/FiO_2_ ratio represented a higher initial disease severity. This finding was in line with the WHO guidelines, in which the extent of the oxygen demand is one of the major criteria for the classification of the disease severity [27]. Nabahati et al. [10] adapted this definition and reported that patients with severe pneumonia (oxygen saturation < 94%, PaO_2_/FiO_2_ < 300, respiratory rate > 30, or lung filtrates > 50%) had a higher risk of pulmonary fibrosis. In a large Chinese cohort study, patients with a high disease severity (high-flow nasal cannula, non-invasive ventilation, or invasive mechanical ventilation) in the acute phase were likely to have a decreased diffusion capacity and CT abnormalities after discharge [28]. This suggests that an excessive viral activity is correlated with a fibroproliferative response and an impairment in the gas exchange.

In previous studies, laboratory findings (including interleukin-6 (IL-6), LDH, interferon-gamma, and KL-6 levels) have been suggested to be prognostic factors for pulmonary fibrosis [9,14,15,29,30]. Our study also attempted to identify the clinically significant laboratory indicators. High LDH (≥400 U/L), AST (≥50 U/L), KL-6 (≥450 U/mL), and low albumin (<3.8 g/dL) levels appeared to be significant in the univariate analysis (*p* < 0.05); however, their predictive values were not proven in our multivariate model. Earlier studies have suggested that high KL-6 is a predictive indicator of pulmonary fibrosis in COVID-19 [14,15]. Although not statistically significant, our results in the multivariate analysis were similar to those of recent studies. Considering that our collection of KL-6 data was incomplete, its predictive power could be promising in future studies.

The percentage of fibrotic change and its components have varied in previous studies exploring the pulmonary consequence of patients with COVID-19 pneumonia, ranging from approximately 20% to 70% [20,31,32,33]. This variance might be derived from the lack of an established clear definition and the occurrence of pulmonary fibrosis following COVID-19. Our study findings were consistent with those of an Egyptian study in which 90 (52.0%) of 173 patients with moderate to severe COVID-19 pneumonia presented with pulmonary fibrosis at a 3-month follow-up. Of the 90 patients, parenchymal bands were observed in 58 (64.4%), bronchiectasis in 11 (12.2%), and honeycombing in 4 (4.4%) [10]. Similarly, in a Chinese study, 141 (68.12%) of 207 patients showed pulmonary fibrosis 91–120 days after the onset [22]. In an Italian study, 72% of 118 patients showed fibrosis-like changes at a 6-month follow-up [31] whereas in another Italian study with 220 patients, radiological sequelae were observed in 45 (20%) patients 3–6 months after discharge [34].

In most studies, patients with more than a moderate disease severity were selected for the study population [10,22,31,34]. Our study also included patients with moderate to severe COVID-19 pneumonia. However, marked differences between the results may have occurred due to varying study designs such as the clinical characteristics of the populations, definitions of fibrosis, and study durations. In real-world clinical practice, it is important to identify patients who are likely to develop fibrotic changes and who require long-term management.

As many patients have reported persistent symptoms after COVID-19, the concept of long COVID has been suggested. Osikomaiya et al. [35] reported that a moderate disease severity had significantly higher odds of persistent COVID-19-like symptoms after discharge than a mild severity. In a systematic review, the most frequent symptom was a shortness of breath or dyspnea, with a median frequency of 36.0% in 26 studies [6]. In clinical settings, respiratory symptoms are often observed in patients post-COVID-19, ranging from minor discomfort to recurrent hospitalization or even death due to an unresolved respiratory failure. 

Fibrotic changes in the lungs have been proposed as the main cause of a reduced pulmonary function. In a 3-month follow-up study conducted by Cocconcelli et al. [34], patients with pulmonary fibrosis were more likely to present with dyspnea with modified Medical Research Council scale scores of 1 and 2. However, a gradual recovery from fibrotic changes and the related symptoms has been observed in long-term studies [22,36]. Moreover, in a British study, an early treatment with corticosteroids was proven to be effective in improving persistent dyspnea and a decreased pulmonary function [37]. Therefore, it is important to understand the progression of pulmonary fibrosis in COVID-19 and identify its prognostic factors from the initial stage of treatment.

The strength of our study was the prospective follow-up of patients with moderate to severe COVID-19. We were able to evaluate the radiological proportion and clinical features of pulmonary fibrosis in patients with a severity requiring hospitalization. In addition, our study identified the clinical risk factors for fibrotic changes, which can be used to predict the disease progression and provide personalized treatment plans.

This study had several limitations. First, the follow-up CT was conducted only once, and 3 months was a relatively short period for evaluating the long-term sequelae of COVID-19. Follow-up CTs for longer period might be needed to provide more evidence for the disease progression and reversibility. Second, this study was conducted at a single center. However, the National Medical Center functioned as a tertiary referral hospital during the COVID-19 outbreak; therefore, our study population may have represented the disease severity and progression of the general population. Third, our study design did not include CT findings at admission. As a result, a quantitative radiologic evaluation of the initial disease severity was absent in the study. Finally, the study design only included patients who successfully recovered and were discharged; those who died during hospitalization were excluded from the study population. Even though our institution was one of the main referral hospitals providing treatment to patients with severe COVID-19, our study population included a small number of patients who required mechanical ventilation or extracorporeal membrane oxygenation. As these populations are expected to present the most severe disease progression, the fibrotic changes in COVID-19 could have been underestimated in this study. Future studies on a large scale with a longitudinal design and a longer follow-up are needed to investigate fibrosis occurring consequently in patients with pneumonia.

## 5. Conclusions

This study revealed that an older age and a low initial SpO_2_/FiO_2_ ratio were crucial in predicting pulmonary fibrosis after COVID-19 pneumonia. Further studies are needed to evaluate the long-term prognosis of fibrotic sequelae and the possible treatment options.

## Figures and Tables

**Figure 1 jcm-11-05913-f001:**
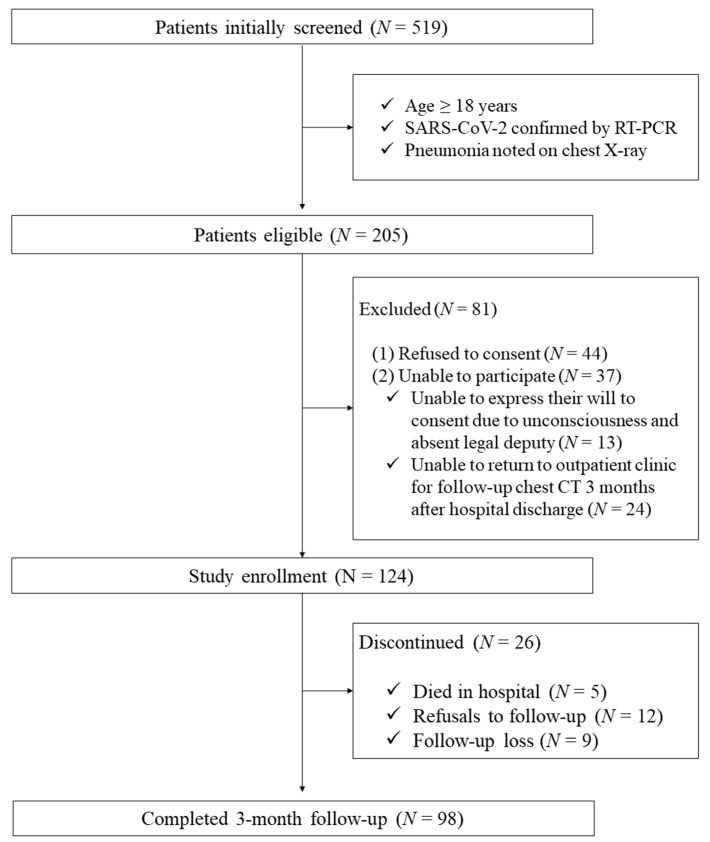
Enrollment flowchart. SARS-CoV-2: severe acute respiratory syndrome coronavirus 2; RT-PCR: reverse transcriptase-polymerase chain reaction; CT: computed tomography.

**Table 1 jcm-11-05913-t001:** Clinical characteristics of the study population.

	Fibrosis	Non-Fibrosis	*p*-Value ^€^
	(*N* = 43)	(*N* = 55)	
Age, years (mean ± SD)	60.7 ± 11.0	50.6 ± 11.5	<0.001
Male (*N,* %)	29 (67.4)	36 (65.5)	0.836
BMI (kg/m^2^) (mean ± SD)	24.5 ± 3.9	26.4 ± 5.2	0.056
Current or ex-smoker (*N*, %)	24 (55.8%)	26 (47.3%)	0.401
Underlying diseases, yes (*N,* %)			
Hypertension	18 (41.9%)	16 (29.1%)	0.188
Diabetes mellitus	10 (23.3%)	12 (21.8%)	0.866
Chronic lung disease *	1 (2.3%)	2 (3.6%)	1.000
Heart disease *	3 (7.0%)	2 (3.6%)	0.651
Stroke *	3 (7.0%)	1 (1.8%)	0.316
Chronic kidney disease *	2 (4.7%)	2 (3.6%)	1.000
Malignancy *	3 (7.0%)	2 (3.6%)	0.651
Charlson Comorbidity Index (mean ± SD)	2.2 ± 1.7	1.2 ± 1.6	0.004
ECOG at admission (mean ± SD)	1.7 ± 1.3	1.3 ± 1.2	0.109
Laboratory finding (mean ± SD)			
CRP (mg/L)	94.9 ± 70.6	75.4 ± 66.4	0.164
LDH (U/L)	465.3 ± 147.0	378.0 ± 188.3	0.016
Albumin (g/dL)	3.6 ± 0.4	4.0 ± 0.4	<0.001
D-dimer (μg/mL FEU)	2.2 ± 4.2	1.2 ± 2.7	0.180
AST (U/L)	65.4 ± 51.1	42.67 ± 23.80	0.009
ALT (U/L)	54.6 ± 54.1	39.7 ± 32.1	0.116
KL-6 (U/mL)	584.9 ± 441.3	345.3 ± 165.0	0.008
BUN (mg/dL)	15.8 ± 8.9	13.9 ± 12.4	0.412
Creatinine (mg/dL)	0.8 ± 0.5	0.8 ± 0.6	0.652
Vital signs (mean ± SD)			
Systolic BP (mmHg)	128.1 ± 21.8	124.9 ± 24.9	0.501
Pulse rate (per minute)	86.5 ± 12.5	88.0 ± 14.4	0.580
Respiratory rate (per minute)	21.7 ± 3.9	21.2 ± 3.5	0.481
Body temperature (Celsius)	37.1 ± 2.8	37.7 ± 0.9	0.104
SpO_2_/FiO_2_, initial	296.0 ± 141.3	390.5 ± 110.3	0.001
Oxygen support, initial, yes (*N*, %) *			0.004
Support was not needed	14 (32.6)	36 (65.5)	
NP or simple mask	17 (39.5)	15 (27.3)	
Reservoir bag or HFNC	11 (25.6)	3 (5.5)	
Mechanical ventilation	1 (2.3)	1 (1.8)	

HFNC: high-flow nasal cannula; ECMO: extracorporeal membrane oxygenation; KL-6: Krebs von den Lungen-6; BMI: body mass index; CRP: C-reactive protein; LDH: lactate dehydrogenase; AST: aspartate aminotransferase; ALT: alanine aminotransferase; BUN: blood urea nitrogen; ECOG: Eastern Cooperative Oncology Group. Chronic lung diseases include asthma, chronic obstructive pulmonary disease, interstitial lung disease, bronchiectasis, TB-destroyed lung, sarcoidosis, and emphysema. Values are presented as a number (%), mean (standard deviation (SD)), or median (interquartile range). ^€^ Tested by chi-squared test for categorical variables and by *t*-test for continuous ones. * Tested by Fisher’s exact test.

**Table 2 jcm-11-05913-t002:** Clinical courses.

	Fibrosis	Non-Fibrosis	*p*-Value ^€^
	(*N* = 43)	(*N* = 55)	
Length of hospital stay, days (mean ± SD)	15.1 ± 8.3	12.7 ± 16.0	0.372
ICU admission, yes (*N*, %)	19 (44.2%)	10 (18.2%)	0.005
Oxygen support, max (*N*, %) *			0.009
Support was not needed	4 (9.3)	13 (23.6)	
NP or simple mask	15 (34.9)	29 (52.7)	
Reservoir bag or HFNC	22 (51.2)	11 (20.0)	
Mechanical ventilation	2 (4.7)	2 (3.7)	
Medication, yes (*N*, %)			
Antiviral agents	37 (86.0)	33 (60.0)	0.005
Monoclonal antibodies	9 (20.9)	8 (14.5)	0.407
Steroids	41 (95.3)	37 (69.8)	0.001
Antibacterial agents	31 (73.8)	32 (60.4)	0.169

HFNC: high-flow nasal cannula; ECMO: extracorporeal membrane oxygenation; KL-6: Krebs von den Lungen-6; NP: nasal prong. Values are presented as a number (%), mean (standard deviation (SD)), or median (interquartile range). ^€^ Tested by chi-squared test for categorical variables and by *t*-test for continuous ones. * Tested by Fisher’s exact test.

**Table 3 jcm-11-05913-t003:** Multivariate regression analysis for independent risk factors for post-COVID-19 fibrosis.

Variable	Crude OR (95% CI)	*p*-Value	Adjusted OR ^€^ (95% CI)	*p*-Value
Age (years)	1.08 (1.04–1.13)	<0.001	1.12 (1.03–1.21)	<0.001
Sex (male)	0.92 (0.39–2.13)	0.842	1.19 (0.30–4.75)	0.804
BMI (kg/m^2^)				
18.0–22.9	1.00 (ref)	0.465 *	1.00 (ref)	0.355 *
23.0–24.9	0.81 (0.28–2.34)	0.694	1.85 (0.40–8.53)	0.428
≥25.0	0.55 (0.20–1.49)	0.237	0.67 (0.15–3.02)	0.604
Ex-smoker or current smoker	1.41 (0.63–3.14)	0.402	2.38 (0.66–8.59)	0.185
Charlson Comorbidity Index				
0–1	1.00 (ref)		1.00 (ref)	
≥2	5.05 (2.13–11.97)	<0.001	0.79 (0.15–4.28)	0.782
KL-6 (U/mL)				
<450	1.00 (ref)	0.055 *	1.00 (ref)	0.208 *
≥450	3.55 (1.26–10.04)	0.017	3.09 (0.76–12.52)	0.115
Unavailable	1.44 (0.55–3.77)	0.453	0.91 (0.24–3.43)	0.885
LDH (U/L)				
<400	1.00 (ref)	0.043 *	1.00 (ref)	0.620 *
≥400	2.96 (1.26–6.92)	0.013	1.03 (0.23–4.58)	0.975
Unavailable	2.06 (0.27–15.80)	0.489	3.94 (0.25–61.84)	0.329
AST (U/L)				
<50	1.00 (ref)		1.00 (ref)	
≥50	3.12 (1.30–7.49)	0.011	3.04 (0.71–13.07)	0.134
Albumin (g/dL)				
≥3.8	1.00 (ref)		1.00 (ref)	
<3.8	3.14 (1.37–7.22)	0.007	2.96 (0.83–10.52)	0.094
SpO_2_/FiO_2_ ratio				
≥300	1.00 (ref)		1.00 (ref)	
<300	4.60 (1.89–11.22)	0.001	7.17 (1.72–29.91)	0.007

BMI: body mass index; KL-6: Krebs von den Lungen-6; ICU: intensive care unit; OR: odds ratio; CI: confidence interval. * *p* for trend. ^€^ Adjusted for all variables listed on the tables.

**Table 4 jcm-11-05913-t004:** Characteristic radiological findings according to the component of pulmonary fibrosis in chest CT scans.

Image Findings, Yes (*N*, % of Fibrosis Group)	
Parenchymal bands	43 (100%)
Traction bronchiectasis ± volume loss	8 (18.6%)
Reticulation	1 (2.3%)
Honeycombing	2 (4.7%)

## Data Availability

De-identified participant data are available upon reasonable request to the corresponding author (Junghyun Kim) by e-mail for the study protocols with the approval of the institutional review board.

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
