# Peer review of "Prognostic Factors for Pulmonary Fibrosis Following Pneumonia in Patients with COVID-19: A Prospective Study"

_jcm, 2022, doi:10.3390/jcm11195913_

Round 1
Reviewer 1 Report
This study investigated which were the predictors of pulmonary fibrosis after COVID-19 pneumonia identified by CT at a follow-up of at least 3 months. The authors found that older age and low initial SpO2/FiO2 ratio were predictors of fibrosis.
The paper is interesting, anyway, I have two major criticisms.
The first one is that a 3 months follow-up is not sufficient to identify fibrosis. In different papers published its has been demonstrated that there is a reduction of the pulmonary signs of inflammation still 1 year after infection. Probably 6 months would be a better choice since after 6 months the changes in pulmonary CT signs are very limited. (10.1148/radiol.202121174, 10.1148/radiol.220019). I think that only patients with a follow-up 6 months or longer should be included in order to avoid biases.
The second major point that should be clarified by the authors is the following. The patients that showed fibrosis had an higher percentage of ICU admissions, anti-viral and corticosteroids use, demonstrating that the COVID-19 pneumonia was worse in this group. Starting from this data, I think that also a CT gravity score at admission should be used in the model of fibrosis development prediction
Author Response
Response to Comments by Reviewer 1
This study investigated which were the predictors of pulmonary fibrosis after COVID-19 pneumonia identified by CT at a follow-up of at least 3 months. The authors found that older age and low initial SpO2/FiO2 ratio were predictors of fibrosis.
The paper is interesting, anyway, I have two major criticisms.
Comment #1.
The first one is that a 3 months follow-up is not sufficient to identify fibrosis. In different papers published it has been demonstrated that there is a reduction of the pulmonary signs of inflammation still 1 year after infection. Probably 6 months would be a better choice since after 6 months the changes in pulmonary CT signs are very limited. (10.1148/radiol.202121174, 10.1148/radiol.220019). I think that only patients with a follow-up 6 months or longer should be included in order to avoid biases.
[Response] We appreciate reviewer’s comment. While we were planning this study in National Medical Center, COVID-19 patients had hospital visit to outpatient clinic shortly after discharge. We observed in the clinic that many patients who had moderate to severe COVID-19 pneumonia still expressed discomfort. With the assumption that the fibrotic sequelae are related with residual symptoms, we designed this study to be 3-month follow-up to demonstrate the contributing factors of short-term aftereffects. Our goal was to find population who would require medical attention during and shortly after acute stage of COVID-19 pneumonia.
Even though the follow-up period was not enough, there were studies [Nabahati M, et al 2021; Gulati A, et al 2017] in which fibrotic changes were observed at three months after COVID-19 pneumonia. Therefore, we would like to suggest 3-month follow-up to be medically relevant to represent progression of COVID-19 infection. Nevertheless, studies with 6 months or longer follow-ups might be needed to provide evidences for long-term prognosis.
We further described this point in the Limitation section (Page 9, Line 270-273).
Revised manuscript
First, follow-up CT was conducted only once, and 3 months was a relatively short period for evaluating the long-term sequelae of COVID-19. Follow-up CTs for longer period might be needed to provide more evidence for the disease progression and reversibility.
References
- Nabahati M, Ebrahimpour S, Khaleghnejad Tabari R, Mehraeen R. Post-COVID-19 pulmonary fibrosis and its predictive factors: a prospective study. Egyptian Journal of Radiology and Nuclear Medicine. 2021, 52, 1-7.
- Gulati A, Lakhani P. Interstitial lung abnormalities and pulmonary fibrosis in COVID-19 patients: a short-term follow-up case series. Clinical Imaging. 2021, 77, 180-186.
Comment #2.
The second major point that should be clarified by the authors is the following. The patients that showed fibrosis had a higher percentage of ICU admissions, anti-viral and corticosteroids use, demonstrating that the COVID-19 pneumonia was worse in this group. Starting from this data, I think that also a CT gravity score at admission should be used in the model of fibrosis development prediction
[Response] Agreeing with reviewer’s comment, we attempted to collect initial CT findings from every patient; however, we were often unable to collect CT scans from patients who were too severe to take CT scan at the right time of (ICU) admission. Even if delayed CT was taken from some patients after stabilization, progression of fibrosis already happened, making accurate analysis difficult.
Usually, it is difficult to have CT scans performed for every COVID-19 patients. COVID-19 guideline recommends chest imaging for initial evaluation, but does not require CT scan for all patients [NIH 2022; WHO 2020]. Clinicians can choose an imaging modality among chest radiography, CT, and lung ultrasound, based on patient’s clinical presentation and resource availability. Therefore, in this study, we decided to evaluate chest X-ray for initial image.
We admit, by collecting admission X-ray instead of CT, we could not present quantitative CT scoring to grade disease severity and predict fibrosis development.
Therefore, we further described this point in the Limitation section (Page 9, Line 276-278).
Revised manuscript
Third, our study design did not include CT findings at admission. As a result, quantitative radiologic evaluation of initial disease severity was absent in the study.
References
- COVID-19 Treatment Guidelines Panel. Coronavirus Disease 2019 (COVID-19) Treatment Guidelines. National Institutes of Health. Available at https://www.covid19treatmentguidelines.nih.gov/. Accessed [2022.9.22].
- Use of chest imaging in COVID-19: a rapid advice guide. World Health Organization; 2020.

Reviewer 2 Report
It was a pleasure to review this article.
My congratulations to the authors for their work. It is clear, easy to read and understand.
I just wanted to make an observation: why didn't was consider calculating and presenting the kappa index? - for agreement between observers (radiologists). It would be interesting information, even for discussion.
Author Response
Response to Comments by Reviewer 2
It was a pleasure to review this article.
My congratulations to the authors for their work. It is clear, easy to read and understand.
I just wanted to make an observation: why didn't was consider calculating and presenting the kappa index? - for agreement between observers (radiologists). It would be interesting information, even for discussion.
[Response] We appreciate for reviewer’s comment. During study period, there were plenty of discussion between the two radiologists, regarding definition and extent of radiologic presentations. However, we attempted to focus on patients’ clinical characteristics and hospital course, and tried to avoid overly focusing on radiologic interpretation. As a result, most decisions on fibrosis based on radiological imaging were consistent with a value of inter-rater Kappa 0.938.
Below we attach radiologic features of three patients in which disagreement occurred between two radiologists. Radiologists usually presented different opinions regarding the presence of parenchymal bands.
|
Patient No. |
Radiologist |
Fibrosis |
Parenchymal band |
Traction bronchiectasis ± Volume loss |
Reticulation |
Honeycombing |
|
25 |
A |
X |
X |
X |
X |
X |
|
B |
O |
O |
X |
X |
X |
|
|
73 |
A |
O |
O |
X |
X |
X |
|
B |
X |
X |
X |
X |
X |
|
|
94 |
A |
O |
O |
X |
X |
X |
|
B |
X |
X |
X |
X |
X |
As reviewer’s comment, we calculated and presented kappa index in the Materials and Methods section (Page 4, Line 118-121).
Revised manuscript
In case of discrepancy between the assessments of the two radiologists, a final consensus was reached after discussion. Inter-rater reliability between radiologists was substantial, with Cohen’s kappa of 0.938.

Reviewer 3 Report
In this commentary, " Prognostic Factors for Pulmonary Fibrosis Following
Pneumonia in Patients with COVID-19: A Prospective Study '' by Inhan Lee et al.
In this commentary, even though including recent and relevant literature, could be
structured more clearly and follow a line of thought.
1. Radiological changes following COVID-19 pneumonia do not resolve completely
in a large minority of patients. Some studies have started to use CT to assess the
presence of long-term lung abnormalities. However, a recent work evaluated 41
patients and showed that in most patients, the chest CT lesions were no longer
present at 7 months after discharge, whereas older patients with severe
comorbidities were more prone to develop fibrosis. It could be useful to add more
about it.
2. It remains uncertain whether the fibrotic-like changes that authors observed
represent irreversible pulmonary fibrosis? and further monitoring is warranted to
answer this question.
3. How should the author treat COVID-19-related end stage lung disease? Just give
explanation.
4. Currently, there is no consensus on the use of antifibrotics in patients with
COVID19-related end stage lung disease. Only two drugs (pirfenidone and
nintedanib) are used to treat idiopathic pulmonary fibrosis (IPF). Both drugs have
been approved by both the European Medicines Agency (EMA) and the United
States Food and Drug Administration (FDA) and can decrease the rate of
pulmonary fibrosis progression. Which type of drug for COVID-19 patients
applied?
Author Response
Response to Comments by Reviewer 3
In this commentary, even though including recent and relevant literature, could be structured more clearly and follow a line of thought.
Comment #1.
Radiological changes following COVID-19 pneumonia do not resolve completely in a large minority of patients. Some studies have started to use CT to assess the presence of long-term lung abnormalities. However, a recent work evaluated 41 patients and showed that in most patients, the chest CT lesions were no longer present at 7 months after discharge, whereas older patients with severe comorbidities were more prone to develop fibrosis. It could be useful to add more about it.
[Response] We appreciate for reviewer’s comment. Consistent with the message of our study that more medical attention should be drawn to older age who are likely to have fibrotic change, we added detailed information in the Discussion section (Page 8, Line 199-202).
Revised manuscript
Older age is a widely acknowledged risk factor for pulmonary fibrosis after COVID-19 in many multivariate analyses (20-22). In a Chinese study, while 61 percent of study population presented complete radiological resolution after seven months, older patients were more likely to develop fibrosis (23).
References
- Liu M, Lv F, Huang Y, Xiao K. Follow-up study of the chest CT characteristics of COVID-19 survivors seven months after recovery. Frontiers in Medicine. 2021, 8, 636298.
Comment #2.
It remains uncertain whether the fibrotic-like changes that authors observed represent irreversible pulmonary fibrosis? and further monitoring is warranted to answer this question.
[Response] As reviewer’s comment, 3-month follow-up is relatively short to evaluate whether the fibrotic change will be reversible or irreversible in the future. Some studies revealed that large percentage of fibrotic change is reversible and typically resolves after 1-year period [Bocchino R et al 2022]. However, a recent study also presents that in certain population, such as old age, fibrotic sequelae may remain longer [Liu M et al 2021].
Unfortunately, longer follow-up was not included in our study design, because our aim was to find clinical characteristics which may help clinicians to identify patients who need extra attention to minimize disease progression. Agreeing to reviewer’s comment, further monitoring is needed to make in-depth and long-term evaluation for study population to elucidate the characteristic of fibrosis that were observed in COVID19 pneumonia patients.
We presented this point in the Limitation section (Page 9, Line 270-273).
Revised manuscript
First, follow-up CT was conducted only once, and 3 months was a relatively short period for evaluating the long-term sequelae of COVID-19. Follow-up CTs for longer period might be needed to provide more evidence for the disease progression and reversibility.
Reference
- Bocchino, R. Lieto, F. Romano, G. Sica, G. Bocchini, E. Muto, et al. Chest CT-based Assessment of 1-year Outcomes after Moderate COVID-19 Pneumonia. Radiology 2022 Pages 220019. Accession Number: 35536134 DOI: 10.1148/radiol.220019
- 2. Liu M, Lv F, Huang Y, Xiao K. Follow-up study of the chest CT characteristics of COVID-19 survivors seven months after recovery. Frontiers in Medicine. 2021, 8, 636298.
Comment #3.
How should the author treat COVID-19-related end stage lung disease? Just give explanation.
[Response] Since the COVID-19 pandemic, practical guidelines recommend corticosteroids in critical hospitalized patients [Bhimrhj A et al 2020; Myall KJ et al 2021]. Studies including case reports have suggested that prolonged, low-dose corticosteroid therapy was also effective for treating post-COVID fibrosis [Kostorz-Nosal S et al 2021].
Contrarily, evidence for prescribing antifibrotics such as pirfenidone and nintedanib is still controversial, and a number of randomized clinical trials are still on progress (ClinicalTrials.gov identifier: NCT04541680, NCT04619680, and NCT04607928).
In our opinion, we try to attempt to prolong steroid therapy to treat COVID-19 related end-stage lung diseases. Further evidence about how long maintain steroids to patients with COVID-19 related end-stage lung diseases should be followed.
References
- Bhimraj A, Morgan RL, Shumaker AH, Baden L, Cheng VC, Edwards KM, Gallagher JC, Gandhi RT, Muller WJ, Nakamura MM, O'Horo JC, Shafer RW, Shoham S, Murad MH, Mustafa RA, Sultan S, Falck-Ytter Y. Infectious Diseases Society of America Guidelines on the Treatment and Management of Patients with COVID-19. Infectious Diseases Society of America. 2022.
- Myall KJ, Mukherjee B, Castanheira AM, Lam, JL, Benedetti G, Mak SM, Preston R, Thillai M, Dewar A, Molyneaux PL. Persistent post–COVID-19 interstitial lung disease. An observational study of corticosteroid treatment. Annals of the American Thoracic Society. 2021, 18, 799-806.
- S. Kostorz-Nosal, D. Jastrzębski, M. Chyra, P. Kubicki, M. Zieliński and D. Ziora.
A prolonged steroid therapy may be beneficial in some patients after the COVID-19 pneumonia. Eur Clin Respir J 2021, 8, 1945186
Comment #4.
Currently, there is no consensus on the use of antifibrotics in patients with COVID19-related end stage lung disease. Only two drugs (pirfenidone and nintedanib) are used to treat idiopathic pulmonary fibrosis (IPF). Both drugs have been approved by both the European Medicines Agency (EMA) and the United States Food and Drug Administration (FDA) and can decrease the rate of pulmonary fibrosis progression. Which type of drug for COVID-19 patients applied?
[Response] As reviewer’s comment, antifibrotic agents are not usually recommended to patients with COVID-19-related lung diseases. Until recently, none of the clinical trials has clearly proven the evidence of treating post-COVID pulmonary fibrosis with antifibrotic agents [Bazdyrev E et al 2021].
In our study, drugs applied for COVID-19 patients were antiviral agents (remdesivir), monoclonal antibodies (regdanvimab, tocilizumab), and steroid. Antibacterial agents were additionally prescribed for patients who were suspected to have coexistent bacterial pneumonia. During the study period, prescribing antifibrotics for COVID-19 pneumonia was not approved by Korean Food and Drug Administration (FDA).
Authors do believe that further researches will give a clue for the use of antifibrotic agents in patients with COVID19-related lung diseases, which do not have clear evidence until now. We clarified this point in the Materials and Methods section (Page 3, Line 100-104).
References
- Bazdyrev E, Rusina P, Panova M, Novikov F, Grishagin I, Nebolsin V. Lung fibrosis after COVID-19: treatment prospects. Pharmaceuticals. 2021, 14, 807.
Revised manuscript
Medications included antiviral agents (remdesivir), monoclonal antibodies (regdanvimab, tocilizumab), steroid, and antibacterial agents. Antifibrotic agents, which did not have clear evidence of use (13) and were not approved by Korean Food and Drug Administration (FDA), were not applied to the study population.
